# Molecular Profiling of A549 Cell-Derived Exosomes: Proteomic, miRNA, and Interactome Analysis for Identifying Potential Key Regulators in Lung Cancer

**DOI:** 10.3390/cancers16244123

**Published:** 2024-12-10

**Authors:** Alexandros Giannopoulos-Dimitriou, Aikaterini Saiti, Andigoni Malousi, Athanasios K. Anagnostopoulos, Giannis Vatsellas, Passant M. Al-Maghrabi, Anette Müllertz, Dimitrios G. Fatouros, Ioannis S. Vizirianakis

**Affiliations:** 1Laboratory of Pharmacology, School of Pharmacy, Aristotle University of Thessaloniki, 54124 Thessaloniki, Greece; gianalex@pharm.auth.gr (A.G.-D.); skaikater@pharm.auth.gr (A.S.); 2Laboratory of Biological Chemistry, Medical School, Aristotle University of Thessaloniki, 54124 Thessaloniki, Greece; 3Proteomics Research Unit, Center of Basic Research II, Biomedical Research Foundation of the Academy of Athens, 11527 Athens, Greece; atanagnost@bioacademy.gr; 4Greek Genome Center, Biomedical Research Foundation Academy of Athens, 11527 Athens, Greece; gvatsellas@bioacademy.gr; 5Department of Pharmacy, Faculty of Health and Medical Sciences, University of Copenhagen, Universitetsparken 2, 2100 Copenhagen, Denmark; 6Laboratory of Pharmaceutical Technology, Department of Pharmaceutical Sciences, Aristotle University of Thessaloniki, 54124 Thessaloniki, Greece; dfatouro@pharm.auth.gr; 7Department of Health Sciences, School of Life and Health Sciences, University of Nicosia, Nicosia 2417, Cyprus

**Keywords:** exosomes, lung adenocarcinoma A549 cells, exosomal proteomic profiling, exosomal miRNA profiling, molecular interactome, bioinformatic analysis, proteogenomics

## Abstract

Exosomes are small extracellular vesicles released by all cells that play a crucial role in cell-to-cell communication. Exosomes derived from cancer cells carry tumorigenic properties, transmitting these signals to surrounding cells within the tumor microenvironment or to distant sites. However, the complexity of exosome cargo remains poorly understood. In this study, we isolated exosomes from human lung adenocarcinoma A549 cell cultures, characterized their physicochemical and morphological properties, and analyzed their proteomic and miRNA cargo through high-throughput profiling and bioinformatics. Key molecules within the exosomal cargo were identified, and network analysis revealed their potential effects on the physiology of potential recipient cells. Furthermore, we compared the miRNA profiles of malignant A549 exosomes with those from normal lung fibroblast MRC-5 cells, identifying tumor-associated miRNAs for potential biomarker exploration in clinical samples. This study advances our understanding of exosomal molecular components and their interactions, opening avenues for further research into their roles in cancer.

## 1. Introduction

Cancer remains a significant public health challenge in the 21st century, with the World Health Organization (WHO) reporting that cancer causes the death of approximately one in twelve women and one in nine men globally [1]. Lung cancer, primarily categorized into small-cell lung carcinoma (SCLC) and non-small lung carcinoma (NSCLC), was the most prevalent malignancy in 2022, accounting for a substantial portion of cancer-related deaths globally [2,3]. Despite recent advancements in understanding the molecular underpinnings of lung cancer and the development of targeted therapies, there is still an urgent need to further elucidate the tumor microenvironment (TME) and the complex systemic networks involved in intercellular communication [4].

Exosomes are small, nano-sized membrane vesicles secreted by almost all living cells, found in various human biofluids, including blood, saliva, and urine. Cancer cells secrete exosomes in greater quantities than normal cells, participating in a well-orchestrated intracellular communication system. Tumor progression strongly depends on neoangiogenesis to support tumor growth, with exosomal cargo affecting multiple signaling pathways, including MAPK, YAP, and VEGF, that regulate angiogenesis [5]. Additionally, exosomes derived from primary tumors can also convey systemic signals that prepare metastatic sites [6]. In addition, tumor cell-derived exosomes can induce the differentiation of normal stromal fibroblasts towards cancer-associated fibroblasts (CAFs), which are crucial for supporting and maintaining tumor development [7]. In this dynamic interplay, CAFs secrete extracellular vesicles that promote epithelial–mesenchymal transition (EMT), proliferation, invasion and migration abilities of tumor cells, aligning with cancer hallmarks [8].

The complexity associated with the molecular cargo of tumor-derived exosomes necessitates the integration of multi-omics high-throughput technologies to identify diverse molecules enriched in exosomes and to apply advanced bioinformatics approaches for discerning molecular targets and their intermolecular interactions involved in tumor angiogenesis, growth, migration, and invasion. Recent advancements towards standardizing exosome isolation methodologies and breakthroughs in next-generation sequencing (NGS) technologies for high-throughput detection and quantification facilitate the identification of exosome-based molecules as diagnostic and prognostic biomarkers for several tumor types, including NSCLC [9]. Previous studies have shown that the exosomal proteome and miRNAome derived from blood liquid biopsies of lung cancer patients can efficiently discriminate them from healthy individuals [10,11,12]. Notably, some exosomal proteins or miRNAs suggest promise as potential biomarkers for diagnosis, prognosis, and anti-tumor drug-resistance and immunotherapy-response prediction [11,13,14,15]. Before testing clinical samples, the isolation of exosomes from in vitro cell cultures of well-established tumorigenic cell lines combined with comprehensive molecular cargo characterization and the integration of proteogenomics and in-depth bioinformatics approaches can lay the foundation for decoding the complex exosomal cargo interactome and elucidate the role of cancer-related exosomes in tumor pathogenesis, progression, migration, invasion, and metastasis.

In this study, we isolated exosomes from the lung adenocarcinoma A549 cell line using a previously established protocol, followed by a detailed characterization of their physicochemical and morphological properties. Common exosomal protein markers, such as CD9 and CD63, were identified in A549 cell-derived exosomes through Western blot and flow cytometry. The isolated exosomes underwent global proteomic and microRNA profiling using NGS-based and nano-LC Mass Spectrometry-based high-throughput methodologies, accompanied by functional bioinformatics analysis to identify cellular compartments, biological processes, molecular functions, and pathways associated with the exosomal cargo. Protein–protein interaction networks were also constructed to identify subnetworks of key proteins that interact with multiple proteins, potentially playing a dominant regulatory role in the exosomal protein interactome.

Following a proteogenomic approach, the target genes of the identified exosomal miRNAs and the protein interactors of the identified exosomal proteins were integrated in complex networks to decipher the exosomal interactome and identify potentially dually affected genes (DAGs) in the potential exosome-recipient cells, potentially deregulated through synergistic or antagonistic interactions. A differential expression and secretion analysis of exosomal miRNAs from A549 cells compared to normal lung MRC-5 cells was performed to identify the upregulated miRNAs in A549 cell-derived exosomes, serving as the foundation for future biomarker-based clinical studies. The schematic representation of the molecular profiling pipeline is shown in Figure 1.

Overall, this study integrates high-throughput discovery data from well-characterized lung adenocarcinoma A549 cell-derived exosomes with novel bioinformatics approaches to elucidate the exosome interactome and the cellular mechanisms impacted by tumorigenic exosomes. We also propose candidate exosomal protein and miRNA biomarkers with strong associations with lung cancer tumorigenicity with potential translatability towards further clinical testing in lung adenocarcinoma patient diagnosis.

## 2. Materials and Methods

### 2.1. Cell Culturing

The human lung adenocarcinoma epithelial A549 (RRID: CVCL_0023) and the human normal fetal lung fibroblasts MRC-5 (RRID: CVCL_0440) cell lines were used for the in vitro assays as exosome-donor cells. These cell lines are routinely maintained and handled in our laboratory following established cell culturing guidelines [16,17]. Specifically, the cells were maintained in a humidified incubator with 5% CO_2_ at 37 °C and cultured in Dulbecco’s modified Eagle’s medium (Thermo Fisher Scientific Inc., Waltham, MA, USA) supplemented with 10% fetal bovine serum (Thermo Fisher Scientific Inc, Waltham, MA, USA) and 1% Penicillin–Streptomycin (Thermo Fisher Scientific Inc., Waltham, MA, USA). The cells were examined morphologically daily for possible contaminations and sub-cultured every 2–3 days at a confluency of approximately 70%. For all the biological assays, including the exosome isolation experiments, only cells with low passage (<15) number were used to ensure their closer resemblance to the original cells and avoid senescence-induced molecular alterations or phenotypic or genotypic changes linked to passage number variations.

### 2.2. Exosome Isolation via Modified Protocol

The exosomes were isolated from the cell culture medium, combining successive low-speed centrifugation steps with sequential ultrafiltration and membrane-based affinity binding [exoEasy Maxi kit (Qiagen, Venlo, The Netherlands)], as reported elsewhere [18]. Particularly, 1.5 × 10^6^ cells were seeded in 100 mm Nunc™ EasYDish™ cell culture dishes (Thermo Fisher Scientific Inc., Waltham, MA, USA) bearing 56.7 cm^2^ surface area. After 24 h, the cell growth medium was removed and the cells were washed twice with PBS 1X, pH 7.4 (Thermo Fisher Scientific Inc., Waltham, MA, USA), followed by the addition of cell growth Dulbecco’s modified Eagle’s medium supplemented with 10% exosome-depleted fetal bovine serum (exosome-depleted FBS, Capricorn Scientific GmbH, Ebsdorfergrund, Germany) and 1% Penicillin–Streptomycin (Thermo Fisher Scientific Inc., Waltham, MA, USA). The exosome-depleted FBS was selected to ensure the absence of any “contaminating” FBS-associated vesicles that could lead to misleading exosome characterization and profiling data. After 24 h and at cell confluency of approximately 70–80%, the cell culture supernatants were collected and subjected to differential low-speed centrifugation steps (2500× *g* for 15 min, 3500× *g* for 15 min, 25,000× *g* for 25 min) to efficiently remove viable cells, dead cells, cell debris as well as the apoptotic bodies and microvesicles as a pellet (EP-medium generation). The supernatant was collected and subjected to sequential ultrafiltration steps using 50 kDa MWCO Amicon^®^ Ultra-15 centrifugal filter units (Merck Millipore, Burlington, MA, USA). After each ultrafiltration step, the concentrated exosome-containing sample was resuspended with PBS 1X, pH 7.4 inside the filter unit and then collected. The concentrated exosome-containing samples were then combined into a single suspension, followed by exosome collection via the membrane-based affinity binding method (exoEasy Maxi Kit, Qiagen, GmbH, Hilden, Germany), following the guidelines of the kit. The purified exosome containing samples were either directly proceeded to physicochemical analysis or stored to −80 °C for downstream assays.

### 2.3. Physicochemical Analysis of Exosomes

The physicochemical analysis of the exosomal samples were conducted via assessing the size distribution, polydispersity index, and zeta potential of the purified exosomes immediately after isolation using the Instrument Nano ZetaSizer (Malvern Instruments, Worcestershire, UK). The instrument features a 4 mW He-Ne laser operating at a wavelength of 633 nm and its detector for the scattered light was set at a fixed angle of 175° from the incident laser beam. For the assessment of size distribution and zeta potential, the exosomal suspension was further diluted 1:20 with PBS 1X, pH 7.4 and analyzed at a stable temperature of 25 °C. During the size distribution analysis, the DLS instrument measures the Brownian motion of the particles and determines their size using the Stokes–Einstein equation. The polydispersity index (PDI), which indicates the sample heterogeneity, is calculated using the formula PDI = (σ/d)^2^, where σ represents the standard deviation of the size distribution, and d is the average particle diameter. The zeta potential is derived by measuring particle velocity using Laser Doppler Velocimetry (LVD), which reveals its Electrophoretic Mobility. The Electrophoretic Mobility, using the Henry Equation, provides the zeta potential of the sample.

### 2.4. Morphological Analysis of Exosomes via Cryo-TEM

The exosomal samples were freeze-dried and prepared for cryo-TEM analysis using a controlled environment vitrification system (CEVS). A small sample volume (5–10 μL) was applied to a carbon film supported by a copper grid and then blotted with filter paper to create a thin liquid layer on the grid. The grid was rapidly frozen in liquid ethane at −180 °C and then transferred to liquid nitrogen (−196 °C). The exosomal samples were examined using a TEM microscope CM120 BioTWIN Cryo (Philips, Amsterdam, Netherlands) equipped with a post-column energy filter GIF 100 (Gatan Inc., Pleasanton, CA, USA) and an Oxford CT3500 cryo-holder (Gatan Inc., USA) with its workstation. The temperature during analysis was kept at −180 °C, and the acceleration voltage was set to 120 kV. Images were captured with a CCD Gatan 791 camera (Gatan Inc., USA) at approximately 1 μm defocus while maintaining low-dose conditions.

### 2.5. CD9 and CD63 Exosomal Markers Identification with Western Blot and Flow Cytometry Analysis

For the detection of CD9 and CD63 exosomal proteins via Western blot, whole cell lysate and exosomal protein content at equal quantity (30 μg of total protein) were analyzed. Briefly, the whole cell lysate was prepared by A549 cell trypsinization and washing with ice-cold PBS 1X, pH 7.5. The cells were then incubated for 30 min in RIPA 1x lysis buffer (Cell Signaling Technology, Danvers, MA, USA) containing PMSF Protease Inhibitors (Thermo Fisher Scientific Inc., Waltham, MA, USA) on ice. A centrifugation step at 14,000× *g* for 10 min was performed and the supernatant was collected. To prepare the exosome lysate, A549 cell-derived exosomes were incubated with 1X RIPA buffer supplemented with PMSF protease inhibitors for 10 min on ice. The protein content of exosomes and whole cell lysate was quantified using the Pierce™ BCA Protein Assay Kit (Thermo Fisher Scientific, Waltham, MA, USA). The protein samples were denaturated at 95 °C for 10 min in Laemmli Lysis-buffer (Merck Millipore, Burlington, MA, USA) and were further separated by electrophoresis in a 15% SDS-PAGE. Color Prestained Protein Standard, Broad Range (New England Biolabs) was used as marker. The proteins were transferred to PVDF Western blotting Membranes (Merck Millipore, Burlington, MA, USA) and the bocking of immunoblots was performed with 1X PBST (0.1% Tween 20) supplemented with 5% skim milk. The immunoblots were then incubated with the appropriate primary monoclonal antibody against CD9 or CD63 exosomal marker (1:5000 dilution of CD9/60232-1-Ig and CD63/67605-1-lg Monoclonal antibodies, Proteintech Group, Rosemont, IL, USA) overnight at RT. The same procedure was also performed for the primary antibody against the β-actin monoclonal antibody (sc-47778, Santa Cruz Biotechnology, Dallas, TX, USA). The next day, the membranes were washed 3 times with 1X PBST (0.1% Tween 20) and incubated with the secondary anti-mouse antibody m-IgG Fc BP-HRP (sc-525409, Santa Cruz Biotechnology, Dallas, TX, USA) for 1h at room RT. Following three additional washing steps, protein bands were visualized using the chemiluminescence Immobilon ECL Ultra Western HRP Substrate (Millipore, Burlington, MA, USA), and a gel imaging system captured images.

CD9 and CD63 protein markers were also detected with flow cytometry using superparamagnetic capture beads conjugated with anti-CD63 antibodies (Abcam Inc., Toronto, ON, Canada). To ensure efficient binding of exosomes on the surface of the capture beads, the XE buffer, used for exosome collection during the final isolation step, was replaced with 1X PBS utilizing the 50 kDa MWCO Amicon^®^ Ultra-15 centrifugal filter units (Merck Millipore, Burlington, MA, USA). Then, 10 μg of purified A549 cell-derived exosomes were incubated overnight at room temperature with 50 μL anti-CD63 capture beads, following the manufacturer’s instructions. The next day, the samples were incubated for 60 min at 4 °C with PE-conjugated anti-CD9 antibody [VJ1/20]. Following multiple washing steps with 1X Assay Buffer, the purified beads conjugated with CD63+ exosomes with either CD9 presence or not were incubated with PE-conjugated mouse monoclonal anti-CD9 antibody. The final product was collected by centrifugation and resuspended in 1X Assay Buffer for flow cytometry analysis.

### 2.6. Proteomic Profiling of Exosomal Proteins via Nano-LC-MS/MS

Exosomes isolated via the modified protocol were freeze-dried for 24 h. The exosome-containing powder was dissolved in 50 μL of lysis buffer (4% *w*/*w* SDS, 0.1 M Tris-HCL, 0.1 M dithioerythritol (DTE) pH = 7.6) and then, probe sonication was performed to the samples following 3 cycles of 38% amplitude and 12 s on/off. The lysis of the samples was performed on the bench for 45 min. After the solubilization of the samples, a centrifugation step at 13,000 rpm for 10 min was performed so that the remaining debris was discarded. According to Bradford assay, 200–500 μg of proteins per sample were placed in the filter unit of the Amicon Ultra 0.5 Centrifugal Filter Device (Merck, Darmstadt, Germany) with 0.1 M DTE in 8 M urea solution and a centrifugation step was performed at 13,000 rpm for 30 min. Then, 0.05 M of iodoacetamide in 8 M urea and 0.1 M Tris-HCL pH = 8.5 were added to the samples and the total reaction was incubated for 20 min at RT in the dark for the reduction and the alkylation of the samples. For the digestion of the samples, a tryptic solution (500 μM) was added for an overnight incubation, with a final trypsin-to-protein ratio of 1:100. The peptide flow-through was freeze-dried, and afterwards, the lyophilized peptide powder was resuspended in 75 μL of HPLC (Thermo Fisher Scientific, Foster City, CA, USA) phase A (99.9% water with 0.1% formic acid (*v*/*v*)). The final peptide mixture was purified utilizing a Millex^®^ syringe-driven filter (Merk KGaA, Athens, Greece) unit and was then analyzed with nano LC-MS/MS. The separation of the peptides was accomplished by a reversed-phase analytical C-18 column (75 μm × 50 cm; 100 Å, 2-μm-bead-packed Acclaim PepMap RSLC, Thermo Scientific) and by using an Ultimate-3000 system (Dionex, Thermo Scientific, Bremen, Germany) interfaced to an LTQ-Velos Orbitrap Elite mass spectrometer (Thermo Scientific, Waltham, MA, USA). Briefly, 6 μL of peptide mixture were loaded on a C-18 pre-column at a constant flow rate of 5 L/min in phase A. The elution time lasted 180 min and had a gradient of 2–35% phase B (99.9% acetonitrile, 0.1% formic acid) at 300 nL/min flow rate. An Orbitrap Elite mass spectrometer fitted with a nanospray source was used to collect the mass spectra. A data-dependent acquisition mode with the XCaliburTM v.2.2 SP1.48 (Themo Scientific, Waltham, MA, USA) software was selected for the instrument. Full scan data were acquired at a resolving power of 60,000, with a maximum integration time of 100 ms. The 20 most intense ions per survey scan were analyzed with the data-dependent tandem mass spectrometry performing higher-energy collision dissociation (HCD) fragmentation in the Orbitrap at a resolving power of 15,000 and a collision energy of 36 NSE%. MS/MS spectra were acquired at a resolving power of 15,000, with a maximum integration time of 120 ms. Measurements were performed using *m*/*z* 445.120025 as lock mass. To minimize the repetitive selection of the same peptide, dynamic exclusion settings were implemented with a repeat duration of 30 s.

### 2.7. MS/MS Data Processing, Protein Abundance Estimation and Bioinformatics Analysis of Proteomic Data

MS/MS data were analyzed with the Proteome Discoverer (version 1.4.0.388, Thermo Scientific, Waltham, MA, USA). The SEQUEST engine against Homo sapiens protein reference database of Uniprot (https://www.uniprot.org/, accessed on 8 June 2022) was selected for the analysis of MS2 spectra and the following Proteome Discoverer parameters were chosen: two maximum missed cleavage points for trypsin; oxidation of methionine as a variable modification; 10 ppm peptide mass tolerance; and 0.05 ppm fragment ion tolerance. A percolator based on q-values at a 0.01% false discovery rate (FDR) validated the peptide spectral matches, and extra peptide filtering was accomplished with an Xcorr versus peptide charge values (percolator maximum Delta Cn was set at 0.05). The doubly and triply charged peptides were analyzed with values of 2.2 and 3.5, respectively. Only peptides with a minimum length of 6 amino acids were accepted.

The corresponding gene identifiers of the proteins isolated from A549 cell-derived exosomes were retrieved by Uniprot’s Retrieve/ID mapping tool (https://www.uniprot.org/id-mapping, accessed on 8 June 2022). To maintain stricter criteria and more robust information on the proteomic data, only the 68 unique proteins identified in at least 2 biological replicates, out of the 252 unique proteins identified in all 3 biological replicates, were used for the downstream bioinformatic analysis.

To determine the relative abundance of each protein with high confidence across the different identified proteins in exosomes, a multi-criteria ranking approach was applied, combining the parameters of Score (confidence of the protein identification), Coverage (percentage of the protein sequence covered by the identified peptides) and Area (integrated peak area of the peptide ion signals). A fixed weight of 0.6 was applied to the metric of Area since this parameter is the most important to estimate the protein relative abundance. The Coverage and Score metrics were assigned with a fixed weight of 0.2. Initially, the values of Area, Score, and Coverage of each protein were normalized to a range between 0 and 1 following this formula:*Normalized Value* = (*Value* − *Min Value*)/(*Max Value* − *Min Value*)
where *Normalized Value* is the Score, Coverage, and Area values of each protein after applying normalization, *Value* is the Score, Coverage, or Area value of each protein, *Min Value* and *Max Value* are the minimum and maximum values of all the proteins of the dataset considering the parameters of Score, Coverage, or Area.

Considering the assigned weights to each metric, the Combined Score of each protein identified in at least 2 of 3 samples was estimated following this formula:*CombinedScore* = (0.2 × *NormalizedScore*) + (0.2 × *NormalizedCoverage*) + (0.6 × *NormalizedArea*)
where *NormalizedScore*, *NormalizedCoverage*, and *NormalizedArea* are the Score, Coverage, or Area of each protein normalized to a range between 0 and 1, and CombinedScore is the final score assigned to each identified protein based on the multi-criteria ranking approach.

The Venn diagram was constructed utilizing the FunRich tool to present the overlaps of the protein identified in A549 cell-derived exosomes with the total exosomal proteomic data registered in Vesiclepedia database [19,20]. In addition, the FunRich tool was used to present the different subcellular locations of the identified proteins according to the cellular compartment in terms of the percentage of proteins found in each compartment with a threshold at log10(*p*-values) > 2 to focus on significantly enriched categories. To identify associations between the identified proteins and those relevant to lung cancer biomarkers, text-mining approaches were applied using DisGeNET with ‘lung cancer’ as the search term, followed by filtering for biomarker relevance [21]. Gene ontology (GO) enrichment as per molecular functions, biological processes, and cellular components, as well as Reactome pathway analysis, were conducted via the clusterProfiler and Reactome PA packages, respectively, in R [22,23]. The protein–protein interaction (PPI) network of the identified proteins was constructed via the STRING database (v11.5), according to the following settings: (a) the minimum interaction score at high confidence (0.7); (b) the max number of interactors shown for the first shell no more than ten; and (c) the max number of interactors shown for the second shell no more than five.

### 2.8. Total microRNA Profiling of Exosomes Using NGS

The total exosomal miRNAs were isolated using the RLT Buffer and the miRNeasy (Qiagen, Limburg, NL, USA) kit according to the manufacturer’s instructions, with slight modifications. Briefly, RLT reagent was added to 0.4 mL of isolated exosomes at a final concentration of 3:1 *v*/*v*, as described elsewhere [24,25]. All samples were homogenized by vortexing for 1 min and incubated for 5 min at RT. Then, 0.4 ml of chloroform was added to each sample (1:1 *v*/*v* chloroform to the starting volume of each sample), and the samples were shaken vigorously for 15 sec and incubated for 3 min at RT. The samples were centrifuged at 12,000× *g* for 15 min at 4 °C. The upper aqueous phase was incubated with 1.5 volumes of 100% ethanol, and the mixture was loaded onto RNeasy mini columns. The column washing steps were performed according to the manufacturer’s instructions.

Small RNA-Seq analysis was performed with the NEBNext^®^ Multiplex Small RNA Library Prep Set for Illumina^®^ (NEB, San Diego, CA, USA). Qubit High Sensitivity DNA assay (Thermo Fisher Scientific, Waltham, MA, USA) was used for the assessment of the small RNA-seq libraries, and the quality control of the libraries was employed with the Agilent Bioanalyzer DNA 1000 kit (Agilent Technologies Inc., Palo Alto, CA, USA). The libraries were combined in equimolar concentrations, and sequencing was carried out by the Illumina NextSeq 2000 sequencer (Illumina, San Diego, CA, USA), generating approximately 20 million single-end reads per sample, with each read measuring 50 bp in length.

### 2.9. microRNA Data Processing and Bioinformatic Analysis

The generated raw sequence reads (20–51 nt length) from the three small RNA-Seq libraries underwent preprocessing to eliminate adapter sequences using Trim Galore! (Galaxy Version 0.6.7 + galaxy0). To enhance the reliability of the analysis, the reads with a base call quality lower than Q25 were excluded and the quality of the remaining reads was evaluated with FastQC (version 0.12.1). Sequences longer than 17 nucleotides following trimming were preserved, while duplicate reads were collapsed using the miRDeep2 Mapper tool (Galaxy Version 2.0.0). The resulting collapsed reads from all samples were mapped to human mature and precursor sequences obtained from miRBase, and the miRDeep2 Quantifier tool (Galaxy Version 2.0.0) was utilized for quantitative expression analysis [26,27]. To enhance the robustness of the analysis, only the miRNAs identified consistently across all three biological replicates were used for further analysis in R. Subcellular localization of the identified miRNAs was conducted with the RNALocate v.2.0 database and functional analysis of the identified miRNAs was conducted using the miRNA Enrichment Analysis and Annotation Tool (miEAA) [26,27]. Accordingly, Gene ontology (GO), Disease ontology (DO), and KEGG pathway enrichment analysis were performed using miRPathDB, miRWalk, and KEGG databases, respectively [28,29,30].

Differential expression analysis of miRNAs identified in cancerous A549 cell-derived exosomes compared to normal MRC-5 cell-derived exosomes was conducted using the DESeq2 bioinformatic tool. Differentially expressed (DE) miRNAs were determined based on a log2 fold change (FC) > 1 or <−1 and an adjusted *p*-value < 0.05 using the Wald test for hypothesis testing in DESeq2.

### 2.10. Proteogenomics-Based Bioinformatics Analysis

To conduct a comprehensive analysis of the biological cargo involved in A549 cell-derived exosomes, a proteogenomic-based bioinformatic analysis was performed, leveraging data from both genomics and proteomics to enhance the interpretation of molecular mechanisms. Briefly, the target genes of the top 20 enriched miRNAs in A549 cells-derived exosomes were retrieved from the miRTarBase [31], the TarBase [32], and miRecords [33] databases using the multiMiR package in R [34]. For enhanced accuracy, the results were further filtered to include only miRNA-gene interactions experimentally validated through methods such as reporter gene assays (e.g., luciferase assays), immunoblotting/Western blotting, and real-time quantitative PCR (RT-qPCR). Accordingly, the protein interactors of the proteins identified in A549 cell-derived exosomes were retrieved by STRING database (v11.5). The circular network of miRNAs, the miRNA target-genes and the exosomal proteins was constructed using Cytoscape (v3.10.1).

## 3. Results

### 3.1. Physicochemical, Morphological, and Biochemical Characterization of A549 Cell-Derived Exosomes, Including CD9 and CD63 Protein Marker Identification

To obtain high purity A549 cell-derived exosomes from cell culture supernatant, free from cell-secretome protein contaminants and FBS-derived proteins, we employed a modified exosome isolation protocol that combines ultrafiltration with a membrane-based affinity method [18], as described previously. DLS analysis revealed exosomes with diameter uniformly distributed at around 30 nm, ranging from 20 to 70 nm (Figure 2a). The polydispersity index (PDI) measured at 0.38 ± 0.014 indicates a heterogenous population of vesicle sizes, which is characteristic of extracellular vesicles [5,35]. Zeta potential analysis of the isolated exosomes diluted in PBS at 4° showed a symmetric, unimodal distribution around −15.5 mV, consistent with other studies evaluating cancer cell-derived exosomes [36,37] (Figure 2b). The cryo-TEM imaging analysis also revealed round-shaped exosomal vesicles (black arrows, Figure 2c–e) with visible lipid-bilayers (white arrows, Figure 2c–e) that had diameters from 25 to 80 nm, supporting the DLS analysis results.

To validate exosome purity and identify common exosomal protein markers from a biochemical perspective, we conducted Western blot and flow cytometry analysis, following MISEV2018 guidelines [38]. Using equal protein amounts from A549 whole cell lysate and A549-cell derived exosomal lysate, we performed a Western blot analysis to detect the tetraspanins CD9 and CD63 [39], along with the negative marker β-actin. The CD9 protein was detected in both cell-lysate and exosomal lysate at approximately 26 kDa (white arrows, Figure 2f), while two CD63 protein zones detected between 40 and 70 kDa (black bracket, Figure 2g), in alignment with previous studies [40]. The additional higher molecular bands for CD9 protein may be attributed to the limited specificity of the antibody, also reported by the manufacturer. Notably, β-actin was only detected in the cell lysate, supporting minimal cytoplasmic contamination in exosomal samples, verifying the efficiency of the proposed exosome isolating approach. 

Flow cytometry analysis using anti-CD63-conjugated superparamagnetic beads also verified the presence of CD9 tetraspanin on the exosome surface. The smaller peak in the histogram’s negative region could indicate beads not conjugated with exosomes, CD63-positive exosomes lacking CD9 expression, or both (Figure 2g).

**Figure 2 cancers-16-04123-f002:**
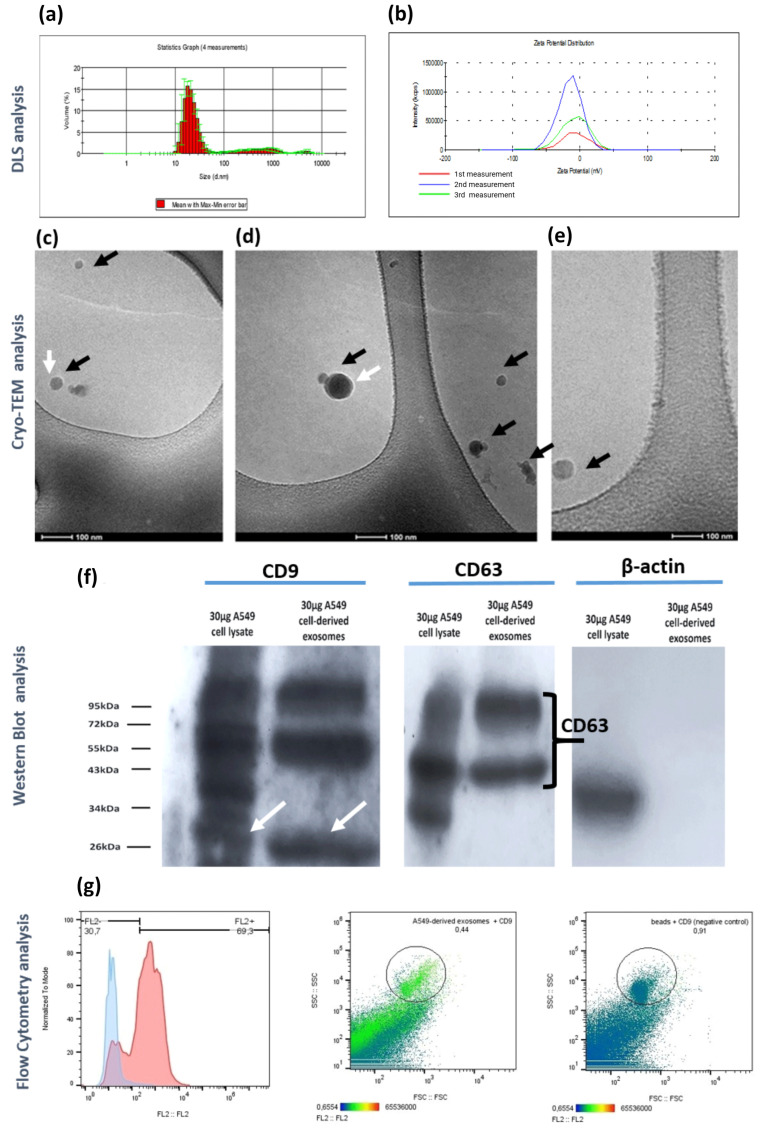
Physicochemical, morphological, and biochemical characterization of A549 cell-derived exosomes concentrated via ultrafiltration and isolated using the membrane-based affinity binding method (exoEasy Maxi Kit, QIAGEN). (**a**) DLS analysis of exosomes considering the parameter of volume. The curve represents the means from three successive DLS measurements (n = 3, biological replicates). (**b**) Representative zeta-potential distribution of the exosomes isolated via the proposed methodology. The curves obtained by three successive measurements (n = 3, technical replicates) are depicted. (**c**–**e**) Cryo-TEM analysis of isolated exosomes. Round-shaped exosomes are depicted (black arrows) with distinct exosomal lipid bilayer (white arrows). (**f**) Detection of CD9 and CD63 exosomal markers as well as cytoplasmic marker β-actin via Western blot analysis. The uncropped bolts are shown in Appendix A. Equal amounts of protein (30 μg) from exosomes and whole cell lysate as calculated by BCA protein assay kit were loaded in wells of 15% SDS-PAGE gel for separation, followed by Western blotting. (**g**) Flow cytometry-based histogram representing the negative control-beads incubated with the anti-CD9 PE antibody without the addition of exosomes (blue curved), as well as the A549 cell-derived exosomes captured by the beads with subsequent incubation with the anti-CD9 antibody (red curve). The dot plots represent the gating strategies for the exosome containing samples and negative control-beads. Data were analyzed using FlowJo (FlowJo™ Software Ashland: Becton, Dickinson and Company, v10.10.0 (2023) [41].

### 3.2. Unraveling the Global Proteomic Landscape of Exosomes Derived from A549 Cancer Cells

The total proteomic analysis of A549 cell-derived exosomes, conducted using nano-LC-MS/MS, identified 68 unique proteins present in at least two biological replicates. Mapping these 68 proteins to their corresponding genes using the FunRich tool [19] resulted in 66 successful protein-to-gene mappings. To estimate the relative abundance of the exosomal proteins, a multi-criteria ranking approach was applied, integrating Score, Coverage, and Area/Intensity. The ten most enriched proteins included PRPF38B (PA: Q5VTL8), ACTB (PA: P60709), ACTG2 (PA: P63267), LUM (PA: P51884), HBA1 (PA: P69905), PRSS1 (PA: P07477), ITIH2 (PA: P19823), GOLM1 (PA: Q8NBJ4), TPM4 (PA: P67936), and SPP1 (PA: P10451) (Figure 3a).

Venn diagram analysis indicated a 98% overlap between the identified exosomal proteins and those registered in Vesiclepedia, supporting the reliability of the results (Figure 3b). Cellular component enrichment analysis via the FunRich tool revealed that most of the exosomal proteins were associated with the extracellular region (70%) and exosomal component (50%), followed by lysosomes (33%) (Figure 3c). This distribution validates the efficiency of the exosome isolation approach, minimizing contamination from other cellular material (cell protein secretome), including FBS-derived components. The percentages presented in FunRich represent the total identified proteins in each compartment relative to overall analysis, which may exceed 100% due to overlapping localizations.

To identify potential associations of the identified proteins with potential biomarkers of lung cancer, text-mining methods were employed via the DisGeNET platform [21]. The top candidate exosomal proteins associated with lung cancer, based on the number of publications supporting the association, included SPP1, CD44, ALB, SPARC, CAT, GAPDH, and CADM1 (Figure 3d).

The Gene ontology (GO) analysis of the proteins identified in at least two biological replicates revealed enrichment of biological processes related to muscle construction, extracellular matrix organization, and various metabolic processes, including glycosaminoglycan, aminoglycan, and mucopolysaccharide metabolism (Figure 4a). The GO cellular component analysis showed significant enrichment in the collagen-containing extracellular matrix, blood microparticles, sarcomeres, and myofibrils (Figure 4b). The observed enrichment of proteins in the endoplasmic reticulum (ER) lumen and vesicle lumen aligns with the exosome biogenesis and secretion pathways, involving intraluminal vesicle formation and cargo delivery [42]. The GO analysis based on molecular function highlighted enrichments related with extracellular matrix structural constituent, antioxidant activity, peroxidation, oxidoreductase activity, and collagen binding (Figure 4c). The pathway analysis revealed significant enrichment in pathways related to carbohydrate metabolism, smooth muscle contraction, ECM organization, and ECM proteoglycan pathways (Figure 4d).

A protein–protein interaction network (Figure 4e) constructed using the STRING database revealed five distinct subnetwork clusters: (a) 18 exosomal proteins that do not interact with other exosomal proteins (red dotted circle), (b) the exosomal proteins KRT1, KRT2, KRT10, and PRSS1 that form a subnetwork (orange dotted circle) related with keratins, (c) several exosomal proteins including the “key” proteins ALB, SPARC, CD44, LUM, and ACAN (purple nodes) that interact with multiple proteins (exosomal and non-exosomal) and form a cluster of proteins mainly located in endocytic vesicles, secretory vesicles, extracellular exosomes (green dotted circle), (d) several exosomal proteins including the “key” proteins GAPDH, ACTB, and NPM1 (purple nodes) that interact with several proteins (exosomal and non-exosomal) forming a cluster of proteins related to cytosol, ficolin-1-rich granules and extracellular vesicles, (e) exosomal proteins including the “key” proteins TNNC2 and MYH1 (purple nodes) that interact with several proteins (exosomal and non-exosomal) and establish a cluster of proteins mainly associated with actin cytoskeleton compartments and intracellular membrane-bounded organelles (purple dotted circle).

### 3.3. Comprehensive Profiling of MicroRNAs Encapsulated in Exosomes Derived from A549 Cancer Cells

To identify microRNAs selectively enriched in A549 cell-derived exosomes, high-throughput small RNA sequencing was performed. The in-depth profiling revealed 72 unique miRNAs consistently detected across all three biological replicates (derived from different cell culture batches). Only miRNAs presented in all replicates were included in subsequent bioinformatic analysis to ensure reliability.

The top 20 miRNAs selectively enriched in A549 cell-derived exosomes, ranked by the average normalized counts across samples, included hsa-miR-619-5p, hsa-miR-122-5p, hsa-miR-9901, hsa-miR-7704, hsa-miR-151a-3p, hsa-miR-423-5p, hsa-miR-1246, hsa-miR-21-5p, hsa-let-7i-5p, hsa-miR-100-5p, hsa-miR-27a-5p, hsa-let-7b-5p, hsa-miR-4448, hsa-miR-184, hsa-miR-10400-5p, hsa-miR-320a-3p, hsa-miR-423-3p, hsa-miR-4516, hsa-miR-148a-3p, and hsa-miR-1290 (Figure 5a). The miRNA subcellular localization analysis based on RNALocate database [43] revealed that these miRNAs are primarily found in exosomes, the extracellular region (circulating or enclosed in extracellular vesicles), and microvesicles (Figure 5b).

To explore the functional implications of these exosomal miRNAs, Over-Representation analysis (ORA) was performed based on miRPathDB [28] and miRWalk databases [29]. The Gene ontology (GO) enrichment analysis based on biological processes revealed that the exosomal miRNAs are mainly involved in regulating metabolic processes (including nitrogen compound and RNA metabolism), gene expression, and cellular macromolecule biosynthetic processes (Figure 5c). The GO cellular component enrichment analysis showed that these miRNAs are mainly localized in the nuclear compartment (nuclear lumen, nucleoplasm, nucleolus), intracellular organelles/vesicles, extracellular vesicles/exosomes and cytoplasm/cytoskeleton (Figure 5d). The GO molecular function enrichment analysis revealed the involvement of the identified miRNAs in several cellular functions, including organic cyclic compound binding, protein/enzyme binding, protein kinase binding, and RNA binding (Figure 5e). The pathway enrichment analysis using miRWalk pathways validated that the A549 cell-derived exosomal miRNAs participate in cellular pathways relevant to cancer pathogenesis/progression (Integrated Pancreatic Cancer Pathway, small-cell lung cancer, chronic myeloid leukemia, prostate cancer, Glioma), as well as in pathways deregulated during cancer progression, such as MAPK, Wnt and p53 signaling pathways, DNA damage response pathways, focal adhesion, and cell cycle regulation (Figure 5f).

### 3.4. Unveiling the Dual Impact: Integrative Proteogenomic Analysis Identifies MicroRNA-Targeted Genes Also Affected by Exosomal Protein Interactors

To elucidate the dual impact of miRNAs and proteins enclosed in the A549 cell-derived exosomes towards potential exosome-recipient cells, a combined network was constructed integrating the miRNA-target gene interactions (from miRTarBase, TarBase, and miRecords databases) and protein–protein interaction (from STRING database) (Figure 6a). This proteogenomic landscape highlights key molecules, including exosomal proteins and miRNAs, that may affect recipient cell physiology through synergistic or antagonistic interactions. As shown in Figure 6a, the inner circle (light green color), which illustrates the dually affected genes (DAGs) that are both targeted by exosomal miRNAs and also interact by exosomal proteins, contains the most genes and demonstrates significant complexity. This network, which incorporates multiple levels of molecular information, suggest the occurrence of complex interactions within potential target cells upon exosome internalization, characterized by co-existing synergistic and mainly antagonistic interactions that eventually disrupt homeostasis and enhance tumorigenicity. Notably, 43 out of 77 exosomal proteins are also target-genes of the top 20 enriched miRNAs derived from A549 cell-derived exosomes.

Text mining using the DisGeNET platform [21] identified the dually affected genes (DAGs)—genes targeted by both exosomal miRNAs and interacting with exosomal proteins—that are associated with lung cancer pathophysiological mechanisms. As illustrated in Figure 6b, the top DAGs associated with lung cancer include ERBB2, CD44, SPP1, APOE, SERPINE1, ENO1, SPARC, and CD74, with ERBB2 and CD44 showing the strongest association (32 publications each).

### 3.5. Differential Secretion Analysis Reveals Distinctive Profiles of Exosomal MicroRNAs Released by Cancerous A549 and Normal MRC-5 Cells

A comparative analysis of miRNAs secretion was performed using the exosomes derived from cancerous A549 cells and exosomes derived normal human lung fibroblast MRC-5 cells, which were previously well characterized in terms of physicochemical and morphological properties, as well as miRNA and proteomic cargo [18]. This analysis aimed to elucidate the distinct miRNA profiles of the studied cell lines, as well as detect the differentially expressed miRNAs and associate them with cancer initiation and progression. Principal component analysis (PCA) showed satisfactory discrimination between exosomal samples derived from tumorigenic and non-tumorigenic cells, supporting their suitability for further functional analysis (Figure 7a).

The differential expression analysis revealed 44 upregulated and 40 downregulated miRNAs in A549 cell-derived exosomes compared to exosomes derived from non-tumorigenic MRC-5 cells (*p* < 0.05; fold change ≥ 2.0) (Figure 7b,d). Among these, 26 miRNAs exhibited exceptionally high expression (log2 fold change > 5), with the top five upexpressed miRNAs being the hsa-miR-619-5p, has-miR-9901, hsa-miR-4755-3p, hsa-4671-5p, and hsa-miR-1303 (Figure 7c).

The GO enrichment analysis using miRTarBase revealed that the upregulation in cancerous A549 cell-derived exosomes significantly enriched molecular functions and biological processes associated with regulatory activities (e.g., regulation of protein phosphorylation and GTPase activator activity), proteolytic processes (e.g., proteasome-mediated protein catabolism), DNA repair and genomic stability, and post-transcriptional regulation (e.g., mRNA 3′-UTR binding) (Figure 8a). The exosomal miRNAs are also involved in cellular processes related with transport and localization (e.g., vesicle-mediated transport from the endoplasmic reticulum to the Golgi, Golgi organization). In addition, the analysis also revealed significant enrichment in cellular compartments associated with nucleus (nuclear membrane), cytoskeleton (microtubule, microtubule cytoskeleton organization), cell–cell junction (adherens junction) and spliceosomal complex. The GO enrichment analysis for biological process via the miRPathDB database revealed that the upregulation in A549 cell-derived exosomes is mainly involved in regulating metabolic processes, cell death, growth rate, apoptosis, stem cell differentiation, DNA repair, negative regulation of TGFβ1 production, and organelle organization (Figure 8b).

The pathway enrichment analysis indicated that the upregulated A549 cell-derived exosomal miRNAs are mainly involved in cancer-related molecular pathways such as the Integrated Pancreatic Cancer Pathway, small-cell lung cancer, chronic myeloid leukemia, Glioma, Renal cell carcinoma, as well as pathways related to focal adhesion and Integrin Signaling. In addition, these miRNAs significantly enriched pathways involved in cell cycle regulation of Cytoplasmic Ribosomal Proteins and, p53, FGF, and Wnt signal transduction processes (Figure 8c). According to the Mammalian ncRNA-Disease Repository (MNDR) database, the upregulated miRNAs in A549 cell-derived exosomes are significantly linked to various cancer types including lung, pancreatic, colorectal, breast, hepatocellular cancers, and Alzheimer’s disease (Figure 8d).

The experimentally validated target genes of the top 5 upregulated miRNAs were also retrieved and introduced into a network depicting the number of miRNAs targeting each gene (Figure 8e). This network analysis enables the identification of target genes in potential exosome-recipient cells that may be strongly affected by the upregulated exosomal miRNAs. Based on the network, the target genes could be categorized into three main groups; (a) the first group (also containing four subclusters) contains only the genes targeted by one of the top five miRNAs, (b) the second group (also containing four subclusters) contains the miRNAs targeted by two different miRNAs, and, (c) the third group contains the genes targeted by at least three different miRNAs, including the KIF3A, POLR3A, YIPF4, GMEB1, NEK8, and IPO9 genes. These genes may exhibit significant dysregulation due to the multidimensional effect induced by multi-targeting by several exosomal miRNAs (dashed blue circle, Figure 8e).

## 4. Discussion

Exosomes derived from malignant cells obtain their oncogenic properties and transmit molecular signals to nearby or distant recipient cells. This process facilitates the neoplastic transformation of normal cells or induces the metastatic potential of tumorigenic cells with initially low migration capacity [44,45]. Advancements in high-throughput next-generation methodologies allow for comprehensive proteomic and miRNA profiling of exosomes, providing an in-depth molecular mapping of their cargo and also facilitating the development of diagnostic panels for tumor diagnosis and monitoring [12,46].

In this study, we presented a comprehensive characterization of exosomes derived from adenocarcinoma A549 cell cultures, including an analysis of their biophysical and morphological properties, as well as in-depth profiling of their protein and miRNA cargo. By leveraging high-throughput methodologies and bioinformatics approaches, we provided a detailed molecular profiling of the exosomal cargo incorporating functional and interaction network analysis. Additionally, we integrated the proteomic and miRNA profiling information into proteogenomics networks, offering a more in-depth view of the exosomal cargo. This approach enables the detection of potential molecular effects on exosome-recipient cells by highlighting dually affected genes that are co-regulated through interactions with both exosomal proteins and miRNAs. In addition, the differential secretion analysis of exosomal miRNAs derived from tumorigenic A549 lung adenocarcinoma cells compared to normal lung fibroblast cells (MRC-5) unveiled distinct miRNA profiles.

The exosomes isolated from the A549 cells displayed a round shape with a distinct lipid-bilayer with uniform distribution at around 40 nm, in accordance with previous studies [47,48]. Their small size and distinctive disk-like dish shape are characteristic attributes that differentiate exosomal vesicles from other EVs [49]. The Western blot and flow cytometry analysis verified the presence of tetraspanins CD9 and CD63 at the exosomal vesicles, also shown by previous studies [40]. Notably, Western blot analysis of the A549 parental cell lysate revealed more protein migration patterns than in exosomal lysates, suggesting the presence of multiple glycosylation forms of these proteins and indicating a possible selective enrichment of specific glycosylation forms within exosomes. CD63 and CD9 tetraspanins promote tumor development and determine invasiveness in certain cancer types [50,51,52], while glycosylation, as a post-translational process, has a key role in modulating the expression and molecular function of these membrane proteins [51].

The total proteomic analysis of A549 cell-derived exosomes revealed that the majority of the exosomal proteins enrich the extracellular region, exosomal component, and lysosomes, validating the credibility of the methodology followed. The top 20 most enriched proteins in A549 cell-derived exosomes included PRPF38B (PA: Q5VTL8), ACTB (PA: P60709), ACTG2 (PA: P63267), LUM (PA: P51884), HBA1 (PA: P69905), PRSS1 (PA: P07477), ITIH2 (PA: P19823), GOLM1 (PA: Q8NBJ4), TPM4 (PA: P67936), and SPP1 (PA: P10451). These highly abundant proteins and their association with key biological processes could shed light on their potential implication in tumor development and progression in potential exosome-recipient cells. For instance, the PRPF38B gene, which encodes the Pre-mRNA-splicing factor 38B, is predicted to participate in mRNA splicing via precatalytic spliceosome. Alterations in the cellular abundance of the proteins involved in mRNA splicing process, potentially facilitated by extracellular exosomes, could lead to gene expression dysregulation through alternative splicing events and/or errors in the splicing process, which are associated with disease development including different types of cancer [53,54]. The LUM gene, encoding a member of the small leucine-rich proteoglycan (SLRP) family, is known to regulate ECM organization and is involved in tumor development and progression [55]. According to previous reports, the elevated expression of LUM in lung cancer cells promotes the metastatic potential of lung cancer cells via autocrine regulatory mechanism [56]. Exosomal LUM is also associated with diagnostic and prognostic value in different malignancies including neuroblastoma [57] and gastric cancer [58]. The HBA1 gene, which encodes the human hemoglobin protein alpha 1, has also been identified as a protein biomarker of NSCLC and SCLC, according to previous studies [59,60,61]. Furthermore, the GOLM1 gene, encoding a resident cis-Golgi membrane protein, promote epithelial–mesenchymal transition (EMT) and tumor cell proliferation, activating matrix metalloproteinase-13 (MMP13) signaling events that contribute to NSCLC aggressiveness [62]. Elevated levels of TPM4, a member of the tropomyosin family of actin-binding proteins involved in cytoskeleton organization, are associated with increased cell migration and motility in lung adenocarcinoma cell lines such as A549 and NCI-H1299 [63].

The bioinformatics-based association analysis identified candidate exosomal proteins associated with lung cancer, including SPP1, CD44, ALB, SPARC, CAT, GAPDH, and CADM1. For instance, the SPP1 gene encoding the secreted phosphoprotein 1, an extracellular matrix (ECM) protein with several adhesion receptor binding domains, is associated with lung cancer progression, resistance to therapy and poor prognosis [64,65]. SPP1 protein could also serve as a biomarker for identifying and predicting prognosis in lung adenocarcinoma and other cancers [64]. Similarly, dysregulation and aberrant expression of CD44, a cell-surface glycoprotein involved in cell–cell interactions, contribute to malignancy initiation and progression potentially serving as both biomarker and therapeutic target [66]. Notably, CD44 protein encapsulated in tumor-derived exosomes, which display that inherent enhanced organotropism may play a pivotal role in maintaining cancer progression, metastasis [67,68], and chemoresistance [69]. Elevated CD44 expression has also been observed in exosomes from bladder cancer cells, supporting its utility as a tumor biomarker [70].

The comprehensive profiling of the miRNAs encapsulated in the A549 cell-derived exosomes and the bioinformatics-based subcellular analysis revealed that the identified miRNAs are primarily localized in exosomes, the extracellular region, and microvesicles. The subcellular analysis results of both isolated exosomal proteins and miRNAs verify the consistency and reliability of the methodologies followed to isolate and analyze macromolecules. The GO enrichment analysis of the identified miRNAs revealed that they participate in regulating several biological processes, including metabolism, gene expression, and cellular macromolecule biosynthesis, molecular functions, such as protein/enzyme binding, protein kinase binding, RNA binding, and, molecular pathways associated with cancer progression including MAPK [71], Wnt [72] and p53 signaling [73], DNA damage response [74], focal adhesion [75], and cell cycle regulation [76]. The top 10 miRNAs selectively enriched in A549 cell-derived exosomes, ranked by the average normalized counts across samples, included hsa-miR-619-5p, hsa-miR-122-5p, hsa-miR-9901, hsa-miR-7704, hsa-miR-151a-3p, hsa-miR-423-5p, hsa-miR-1246, hsa-miR-21-5p, hsa-let-7i-5p, and hsa-miR-100-5p. For instance, the tumor derived exosomal hsa-miR-619-5p, which suppresses RCAN1.4 gene expression, has been associated with tumor growth and metastasis and angiogenesis in endothelial HUVEC cells and lung adenocarcinoma A549 cells [77]. Also, the upregulated expression of hsa-miR-619-5p in plasma-derived exosomes of lung cancer patients could facilitate potential biomarkers for the early diagnosis of lung adenocarcinoma, according to previous studies [78]. Many studies have also shown that miR-122-5p is dysregulated in NSCLC cell-derived EVS and the delivery of miR-122-5p miRNAs enclosed in the lung cancer cell-derived EVS-promoted migration of liver cells, establishing a pre-metastatic environment for lung cancer metastasis towards liver [79]. Hsa-miR-122-5p is also involved in NSCLC progression regulating the P53 protein, which is also involved in the lipid metabolism of cancer cells via the mevalonate (MVA) pathway. Inhibiting miR-122-5p expression in A549 cells resulted in cell proliferation and migration reduction [80]. Similarly, exosomal hsa-miR-151a-3p is associated with lung cancer metastasis towards bone potential serving as a novel biomarker for predicting bone metastasis in lung cancer [81]. Similarly, the miRNAs hsa-miR-423-5p, hsa-miR-1246, and miR-21-5p are associated with lung adenocarcinoma aggressive behavior, stemness, and invasiveness, as well as biomarkers for lung cancer prognosis [82,83,84].

In this study, we also employed a proteogenomics approach by integrating miRNA and proteomic data obtained from miRNA sequencing and total proteomics analysis of well-characterized exosomes derived from the tumorigenic A549 cells. This dual analysis facilitates a more in-depth understanding of the molecular interplay between the exosomal cargo and the potential exosome-recipient cells. By incorporating the multi-level molecular information into complex networks that illustrate interactions among the identified exosomal miRNAs with their target genes and the exosomal proteins with their co-interacting proteins, we aimed to identify genes and proteins in exosome-recipient cells within the tumor microenvironment that may be dually influenced, either synergistically or antagonistically, by the exosomal constituents. This methodology provides valuable insights into the regulatory effects of tumor cell-derived exosomes on cellular functions during cancer progression. The network analysis highlighted the complex molecular interactions within potential target cells upon exosome uptake, with both synergistic and antagonistic effects that may disrupt cellular homeostasis and promote tumorigenic effects. For instance, the CD44 protein, reported to mediate metastasis via the a6β4 integrin [68] is targeted by four of the thirteen exosomal miRNAs depicted in the network. Despite being a miRNA target, CD44 was also detected as an exosomal protein, indicating a potential competitive effect since exosomes deliver both suppressive miRNAs and active CD44 protein to modulate recipient cell function. Additionally, 7 out of 13 exosomal miRNAs, amongst them the has-miR-21-5p, target the oncogene SET, known for its role in cell cycle regulation, motility (via the SET-Rac1 complex), and metastasis (through inhibition of NM23H1) [85]. Additionally, proteins of the network such as ACAN, CD44, CD74, EEF2, EIF2S1, and ALB, interact with both exosomal miRNAs including hsa-let-7b-5p, hsa-miR-122-5p, hsa-miR-27a-5p, and hsa-miR-423-5p, and exosomal proteins, which potentially affects their expression and cellular function resulting in dysregulated cellular pathways and imbalanced homeostasis of the exosome-recipient cells.

Implementing a gene-to-disease association analysis, we also found that the top 10 dually affected genes (DAGS) associated with lung cancer are ERBB2, CD44, SPP1, GAPDH, APOE, SERPINE1, ENO1, SPARC, CD74, and SET. These genes play roles in receptor signaling, cell adhesion, extracellular matrix organization, and metabolic regulation. These DAGs could serve as key dysregulated genes of the cells internalizing lung cancer-derived exosomes, potentially resulting in cancer-associated cellular alterations involved in tumor maintenance and growth.

This study compared miRNA expression between tumorigenic A549 and non-tumorigenic MRC-5 cells, chosen for their relevance to lung cancer. The differential expression analysis revealed 44 upregulated and 40 downregulated miRNAs in exosomal vesicles derived from tumor cells compared to those derived from non-tumorigenic cells. The GO functional analysis of the upregulated miRNAs indicated their involvement in regulatory activities such as GTPase activity, DNA repair, apoptosis, cell death, as well as processes related to the cellular transport of macromolecules. MiRNA-disease association analysis linked these miRNAs to various cancers and Alzheimer’s disease, suggesting broader impacts on cellular homeostasis. Based on the analysis, the miRNAs hsa-miR-619-5p, hsa-miR-9901, hsa-miR-4755-3p, hsa-miR-4671-5p, and hsa-miR-1303 exhibited the most significant upregulation in the tumor-derived exosomes. Notably, tumorigenic hsa-miR-619-5p is both the most upregulated miRNA in the differential expression analysis and the most selectively enriched miRNA in A549 cell-derived exosomes, establishing it as a promising target for further biological studies to elucidate its precise role in tumor progression, as well as for clinical studies investigating its potential as a biomarker for disease diagnosis. Previous studies have shown that hsa-miR-1303 displayed elevated expression in NSCLC tissue samples also associated with tumor stage, serving as potential biomarker for disease diagnosis and prognosis prediction [86].

Furthermore, network analysis, which incorporates the target genes of the top five upregulated miRNAs, facilitated the identification of key genes in exosome-recipient cells that may be significantly affected by these miRNAs. This analysis revealed that the KIF3A, POLR3A, YIPF4, GMEB1, NEK8, and IPO9 genes are targeted by at least three different miRNAs. These genes could serve as key molecular players that may experience significant expression alterations in recipient cells following exosome internalization. For instance, the KIF3A gene, which encodes the Kinesin family member 3A motor protein, has been associated with carcinogenesis inhibition and apoptosis induction in NSCLC cells, as well as with adverse survival outcomes in lung cancer patients bearing low KIF3A protein expression levels. Therefore, the targeting of the KIF3A gene by multiple exosomal miRNAs could potentially result in tumorigenic effects.

The study’s exclusive use of A549 cells as a well-established non-small-cell lung cancer in vitro model limits the generalizability of the findings to other lung cancer subtypes or cell lines. Further studies using additional models are required to expand these results and elucidate the exosome-based molecular interactome in lung cancer.

In addition, while this study identifies potential key regulators in exosome-based lung cancer research, further experimental validation is required to confirm their roles in disease progression and pathogenesis. Functional studies, including gene targeted analysis and in vivo models are needed to establish casual relationships. Validation in patient-derived samples could further enhance translational relevance of these key regulator molecules.

## 5. Conclusions

To the best of our knowledge, this is the first study that comprehensively assess the molecular landscape of lung adenocarcinoma A549 cell-derived exosomes, integrating high-throughput proteomics, miRNA sequencing, and exosome interactome networking. By combining this multi-layer information, we gain a more holistic view of the exosomal cargo, revealing complex molecular networks that drive cancer progression. Through proteomic analysis, we identified key proteins that may play crucial roles in mediating the interactions between tumor cells and their microenvironment. Concurrently, miRNA sequencing highlighted specific exosomal RNAs that modulate gene expression and signaling pathways linked to tumorigenesis. Notably, hsa-miR-619-5p emerged as a key regulatory molecule, being the most enriched and differentially expressed miRNA in A549 cell-derived exosomes compared to normal MRC-5 cell exosomes. Furthermore, integrating a proteogenomic perspective allowed us to interpret the functional consequences and dynamics of the identified molecules in recipient cells. In conclusion, this study presents a novel approach to elucidate the exosomal molecular interactome and its potential biological effects on recipient cells during cancer progression.

## Figures and Tables

**Figure 1 cancers-16-04123-f001:**
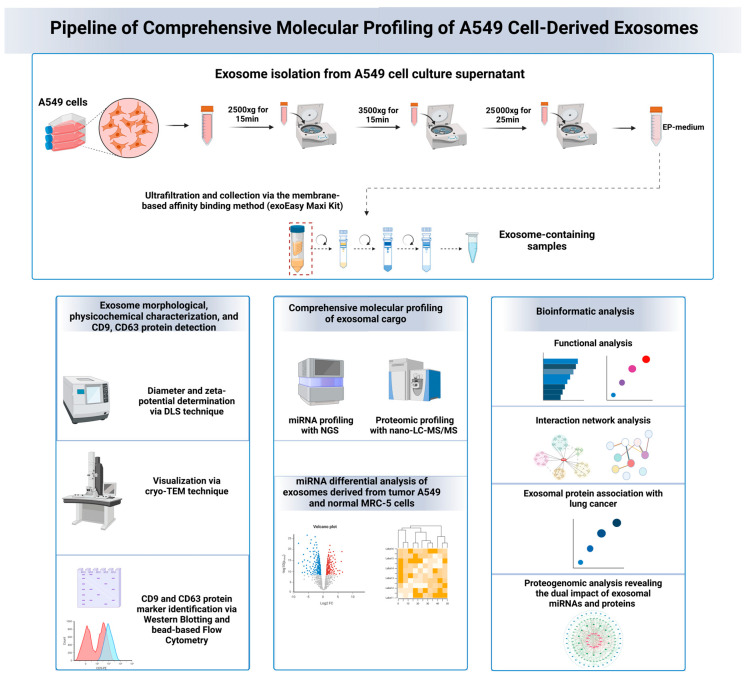
Schematic representation of the molecular profiling pipeline used for the analysis of lung adenocarcinoma A549 cell-derived exosomes. The pipeline includes exosome isolation, characterization (physicochemical and morphological properties), followed by proteomic profiling and miRNA sequencing. Bioinformatic analysis was performed to identify the differentially expressed miRNAs in cancerous A549 cell-derived exosomes compared to normal MRC-5 cells. Functional enrichment and exosome molecular interactome analysis were conducted to explore the effects of exosomal cargo and identify exosomal key regulatory molecules involved in lung cancer progression. Proteogenomic network analysis was also performed to determine key molecules (genes and proteins) that are dually regulated by exosomal miRNAs and proteins through extensive interactions.

**Figure 3 cancers-16-04123-f003:**
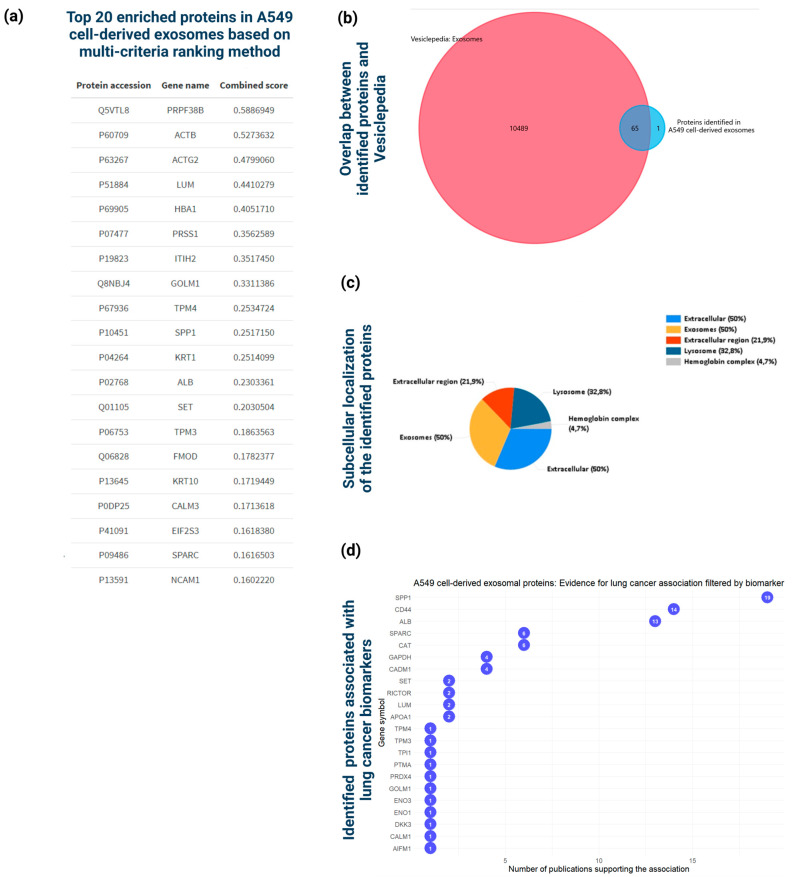
Comprehensive proteomic analysis of the 68 proteins identified in at least two biological replicates of the A549 cell-derived exosomes isolated via the modified protocol. (**a**) Top 20 enriched exosomal proteins identified using a multi-criteria approach, integrating Score (protein identification confidence), Coverage (percentage of protein sequence covered by identified peptides), and Area (integrated peak area of peptide ion signals). (**b**) Venn diagram illustrating protein overlap between the A549 cell-derived exosome and exosomes registered in Vesiclepedia database. The overlap analysis was conducted by the FunRich tool. (**c**) Cellular component enrichment analysis according to the FunRich tool depicting the five different statistically significant (log10(*p*-values) > 2 subcellular localizations of the identified proteins. (**d**) Plot illustrating the association of the identified exosomal proteins with lung cancer biomarkers, derived from text-mining methods and DisGeNET database.

**Figure 4 cancers-16-04123-f004:**
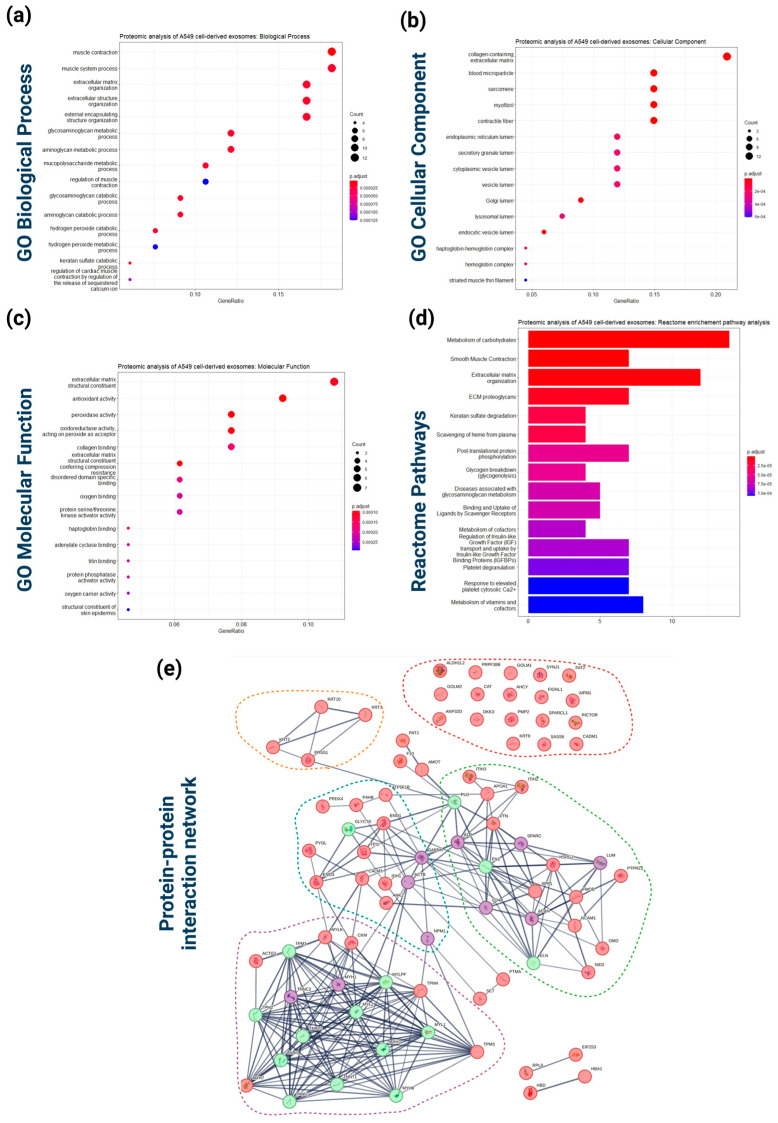
Functional bioinformatics-based analysis of the identified A549 cell-derived exosomal proteins. (**a**) Top 15 significantly enriched GO biological process (BP) terms associated with the genes mapped to the proteins identified in A549 cell-derived exosomes. The gene ratio and statistical significance (*p*-value < 0.05, following Benjamini and Hochberg’s adjustment method) are also depicted. (**b**) Top 15 significantly enriched GO cellular component (CC) terms associated with the genes mapped to the proteins identified in A549 cell-derived exosomes. The gene ratio and statistical significance (*p*-value < 0.05, following Benjamini and Hochberg’s adjustment method) are also depicted. (**c**) Top 15 significantly enriched GO molecular function (MF) terms associated with the genes mapped to the proteins identified in A549 cell-derived exosomes. The gene ratio and statistical significance (*p*-value < 0.05, following Benjamini and Hochberg’s adjustment method) are also depicted. (**d**) The Reactome pathway enrichment analysis on the genes mapped to the proteins identified in A549 cell-derived exosome. The top 15 statistically significant pathways are listed, and their colors correspond to the adjusted *p*-values. (**e**) PPI network showing the interactions amongst the proteins identified in A549 cell-derived exosomes. The network was constructed using the STRING-DB tool. The exosomal proteins are shown as red nodes (low number of linked interacting proteins) and purple nodes (high number of linked interacting proteins). Proteins directly interact with the exosomal proteins and are not identified in exosomes as depicted as greed nodes. The line width indicates the confidence of data support. A minimum confidence interaction of 0.7 was selected for high confidence. The multi-colored dotted lines enclose 5 different subnetworks of exosomal proteins.

**Figure 5 cancers-16-04123-f005:**
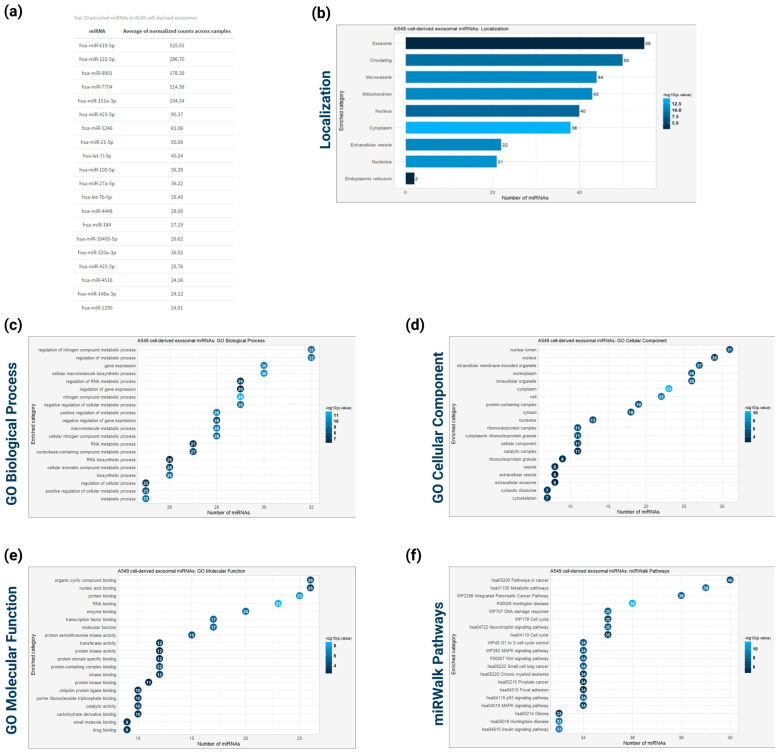
Global microRNA profiling of exosomes derived from A549 cells. (**a**) Top 20 enriched miRNAs based on the average of normalized count across the samples. (**b**) Subcellular localization analysis of the identified miRNAs based on RNALocate database. (**c**) GO enrichment analysis based on Biological Process of the identified exosomal miRNAs. (**d**) GO enrichment analysis based on Cellular Comonent of the identified exosomal miRNAs. (**e**) GO enrichment analysis based on molecular function of the identified exosomal miRNAs. (**f**) Pathway enrichment analysis based on Reactome of the identified exosomal miRNAs. For all enrichment analyses, the Benjamani–Hochberg method was used to adjust *p*-values for false discovery rate (FDR), with a significance threshold set at *p* < 0.05.

**Figure 6 cancers-16-04123-f006:**
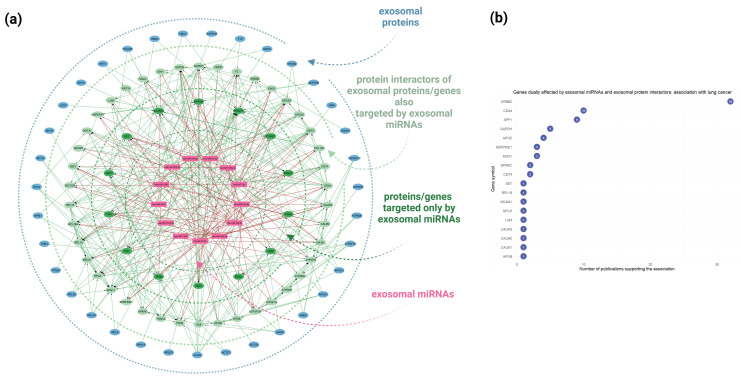
Proteogenomic approach combining multi-level molecular information associated with A549 cell-derived exosomes as obtained via high-throughput methodologies. (**a**) A combined network analysis depicting the dual impact of miRNAs and proteins encapsulated within A549 cell-derived exosomes on potential recipient cells. The network integrates miRNA-target gene interactions (sourced from miRTarBase, TarBase, and miRecords databases) with protein–protein interaction retrieved from the STRING database. Key genes affected by both exosomal miRNAs and proteins, through synergistic or antagonistic interactions, are represented in more complex subnetworks. The blue nodes represent the identified exosomal proteins, the light green nodes indicate the protein interactors of the exosomal proteins/genes that are also targeted by exosomal miRNAs, the intense green nodes represent genes targeted solely by exosomal miRNAs, and the purple nodes denote the top 20 exosomal miRNAs. (**b**) A plot depicting the results of text mining conducted using the DisGeNET platform to identify dually affected genes (DAGs) associated with lung cancer pathophysiological mechanisms.

**Figure 7 cancers-16-04123-f007:**
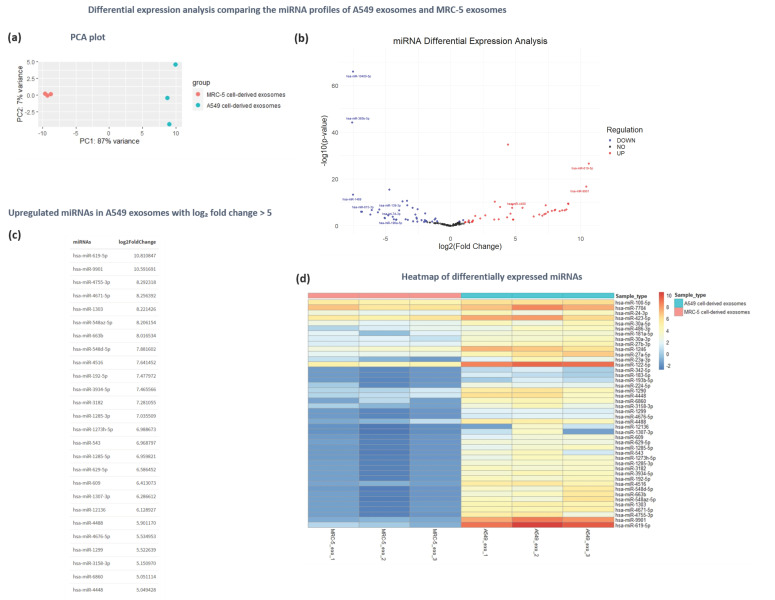
Differential expression analysis of exosomal miRNAs derived from tumorigenic A549 cells and normal MRC-5 cells. (**a**) PCA plot demonstrating the clustering of samples derived from A549 and MRC-5 exosomes based on miRNA expression profiles. (**b**) Volcano plot illustrating the differentially expressed miRNAs between A549 and MRC-5 exosomes, with the x-axis displaying log2(fold change) and the y-axis showing −log10(*p*-value). (**c**) The top upregulated miRNAs in A549 cell-derived exosomes with log2 fold change > 5. (**d**) Heatmap displaying the differentially expressed miRNAs between A549 and MRC-5 exosomes, with miRNAs on the y-axis and samples on the x-axis. Color intensity represents log2(fold change), with warm colors indicating upregulation and cool colors indicating downregulation in A549 cell-derived exosomes.

**Figure 8 cancers-16-04123-f008:**
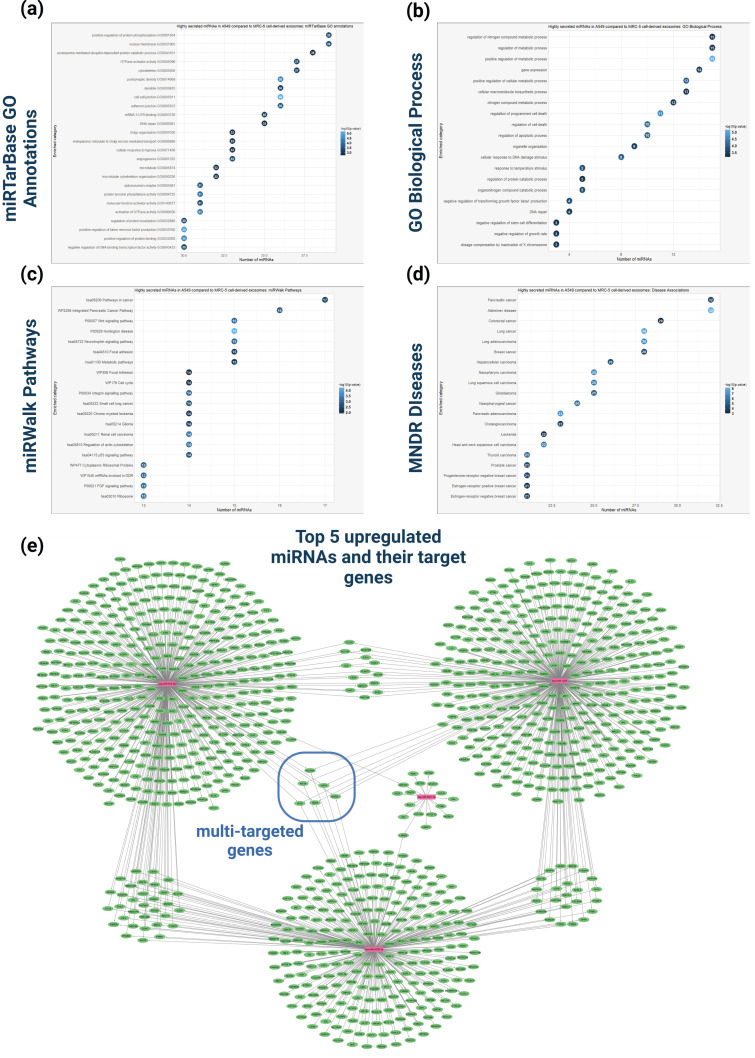
Functional analysis of the upregulated miRNAs in A549-cell derived exosomes compared to MRC-5 cell-derived exosomes. (**a**) GO enrichment analysis of the upregulated miRNAs based on miRTarBase resource. (**b**) GO enrichment analysis based on biological process of the upregulated miRNAs. (**c**) Pathway enrichment analysis based on miRWalk database of the upregulated miRNAs. (**d**) Disease enrichment analysis, based on MNDR (Molecular Network of Disease Relations) database to identify diseases significantly associated with the upregulated miRNAs. (**e**) Network representation of experimentally validated target genes for the top five upregulated miRNAs. The miRNAs and their target genes are represented with purple and green nodes, respectively. Edges between miRNAs and genes illustrate validated interactions. This network visualization underscores potential key regulatory genes associated with the highly expressed miRNAs in the dataset.

## Data Availability

Research data supporting this publication will be made available from the corresponding author on reasonable request.

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
