# Peer review of "Molecular Profiling of A549 Cell-Derived Exosomes: Proteomic, miRNA, and Interactome Analysis for Identifying Potential Key Regulators in Lung Cancer"

_cancers, 2024, doi:10.3390/cancers16244123_

Round 1
Reviewer 1 Report
Comments and Suggestions for Authors
Here, the authors investigate the molecular profiling of exosomes derived from A549 cells, a human lung adenocarcinoma cell line. The researchers employed high-throughput proteomic and miRNA profiling, coupled with bioinformatics network analysis, to elucidate the molecular cargo of these exosomes. A comparative analysis was also conducted with exosomes derived from normal lung fibroblast MRC-5 cells. The article is well written, uses current approaches, and will be interesting for the readership of the journal.
Main points
The study provides a thorough analysis of the proteomic and miRNA content of A549 cell-derived exosomes, offering valuable insights into the molecular mechanisms underlying lung cancer.
The use of bioinformatics tools to analyze the interactome and identify potential key regulators is a significant strength, as it allows for the identification of novel targets for therapeutic intervention.
By comparing the exosomes from cancerous and normal cells, the study highlights specific molecular differences that could be crucial for understanding cancer progression and metastasis.
Minor points
The study focuses solely on A549 cells, which may limit the generalizability of the findings to other types of lung cancer or different cancer cell lines. This should be discussed in a limitations paragraph.
While the study identifies potential key regulators, further experimental validation is required to confirm their roles in lung cancer. This should be discussed in a long term prospects paragraph.
Author Response
|
1. Summary |
|
|
|
Thank you very much for taking the time to review this manuscript. Please find the detailed responses below and the corresponding revisions highlighted in the re-submitted files.
|
||
|
2. Questions for General Evaluation |
Reviewer’s Evaluation |
Response and Revisions |
|
Does the introduction provide sufficient background and include all relevant references? |
Yes |
[Thank you for your evaluation] |
|
Are all the cited references relevant to the research? |
Yes |
|
|
Is the research design appropriate? |
Yes |
|
|
Are the methods adequately described? |
Yes |
|
|
Are the results clearly presented? |
Yes |
|
|
Are the conclusions supported by the results? |
Yes |
|
|
3. Point-by-point response to Comments and Suggestions for Authors |
||
|
Comments 1: [The study focuses solely on A549 cells, which may limit the generalizability of the findings to other types of lung cancer or different cancer cell lines. This should be discussed in a limitations paragraph.]
|
||
|
Response 1: Thank you for your valuable comment regarding the generalizability of our findings. We have addressed this concern by adding a limitations paragraph to the Discussion section (line 860-863, page 27). The paragraph highlights the exclusive use of A549 cells in our study and acknowledges that this may limit the extrapolation of our findings to other lung cancer subtypes or cell lines.
Line 860-863, page 27- We incorporated the following text: “The study’s exclusive use of A549 cells as a well-established non-small cell lung cancer in-vitro model limits the generalizability of the findings to other lung cancer subtypes or cell lines. Further studies using additional models are required to expand these results and elucidate the exosome-based molecular interactome in lung cancer”.
This addition aims to transparently acknowledge the scope of the study and the need for further research, in line with your suggestion.
|
||
|
Comments 2: [While the study identifies potential key regulators, further experimental validation is required to confirm their roles in lung cancer. This should be discussed in a long term prospects paragraph.] |
||
|
Response 2: Thank you for your insightful suggestion regarding the need to discuss further experimental validation of the identified key regulators. In response, we have added a paragraph to the Discussion section (line 864-869, page 27) under a "long-term prospects" context. This paragraph acknowledges the requirement for additional research to confirm the roles of these key regulators in lung cancer.
Line 864-869, page 27- We incorporated the following text: “In addition, while this study identifies potential key regulators in exosome-based lung cancer research, further experimental validation is required to confirm their roles in disease progression and pathogenesis. Functional studies, including gene targeted analysis and in vivo models are needed to establish casual relationships. Validation in patient-derived samples could further enhance translational relevance of these key regulator molecules”.
This addition reflects the importance of future work to solidify the findings and emphasizes potential translational avenues, as you recommended.
|
||
|
4. Response to Comments on the Quality of English Language |
||
|
Point 1: The quality of English does not limit my understanding of the research. |
||
|
Response 1: Thank you for your evaluation. |
||
|
5. Additional clarifications |
||

Reviewer 2 Report
Comments and Suggestions for Authors
Present article titled "Molecular Profiling of A549 Cell-Derived Exosomes: Proteomic, miRNA, and Interactome Analysis towards Identifying Potential Key Regulators in Lung Cancer" is a study on understanding of exosome molecular components and their interactions. I have following suggestions:
1. There is an overlap of methods section with results and Introduction. Please keep the methods section at one place to make to easy to understand.
2. The Figure 2 labelling has typo errors. The figure a, b, c, d, f, h is there in the figure labelling but in the figure legends its not matching. Please correct that.
3. Line 411- Please add reference for Flowjo software.
4. Figure2- Western blot image does not have loading control. Please add a total protein stain image with the blots to show that loading was equal in all samples.
5. Western blot image does not have a loading marker.
6. The figure 5 a miRNA fold change compared to control should be mentioned for representative miRNAs.
7. Few pictures specially enrichment analysis and network analysis are not high quality. Its hard to read them.
Author Response
|
1. Summary |
|
|
|
Thank you very much for taking the time to review this manuscript and for your valuable comments. Please find the detailed responses below and the corresponding/corrections highlighted in the re-submitted files.
|
||
|
2. Questions for General Evaluation |
Reviewer’s Evaluation |
Response and Revisions |
|
Does the introduction provide sufficient background and include all relevant references? |
Can be improved |
[We have thoroughly reviewed the entire manuscript to improve its clarity, incorporating your comments. We carefully checked the citations and minimized overlap between the Methods, Introduction, and Results sections to ensure that the Results section clearly highlights the study’s findings. Additionally, we addressed typographical and grammatical errors and enhanced the quality of the images in line with your suggestions.] |
|
Are all the cited references relevant to the research? |
Yes |
|
|
Is the research design appropriate? |
Yes |
|
|
Are the methods adequately described? |
Yes |
|
|
Are the results clearly presented? |
Can be improved |
|
|
Are the conclusions supported by the results?
|
- |
|
|
3. Point-by-point response to Comments and Suggestions for Authors |
||
|
Comments 1: [There is an overlap of methods section with results and Introduction. Please keep the methods section at one place to make to easy to understand.] |
||
|
Response 1: [ We have thoroughly reviewed the entire manuscript, taking your comments into account, and made efforts to improve its clarity. Specifically, we minimized overlap between the Methods and the Introduction/Results sections by removing repetitions to better focus on the main scope of each section. The following changes have been incorporated into the manuscript: 1.Introduction Section: Page 3, line 106-Further details were deleted to avoid overlaps between Methods and Introduction. 2.Results Section: Page 12, lines 419 & 429- Further details were deleted to avoid overlaps between Methods and Results. Results Section: Page 18, line 548-Further details were deleted to avoid overlaps between Methods and Results. Results Section: Page 23, line 671-Further details were deleted to avoid overlaps between methods and results.]
|
||
|
Comments 2: [The Figure 2 labelling has typo errors. The figure a, b, c, d, f, h is there in the figure labelling but in the figure legends its not matching. Please correct that.] |
||
|
Response 2: [Thank you for your careful evaluation. The image has been revised to include the appropriate labels (a, b, c, d, e, f, g) in alignment with the corresponding legend. The updated image has been re-uploaded.]
Comments 3: [Line 411- Please add reference for Flowjo software.] Response 3: [We have included a comment in the manuscript indicating the appropriate citation to be added, ensuring that the format of the citations remains unchanged.]
Comments 4: [Figure2- Western blot image does not have loading control. Please add a total protein stain image with the blots to show that loading was equal in all samples.] Response 4: [Thank you for your insightful comment regarding the inclusion of a loading control for the Western blot image in Figure 2. While we understand the importance of demonstrating equal loading, the samples in this study include whole-cell lysates and exosomes, which are biologically distinct. Due to the unique protein compositions of exosomes, which lack many of the proteins constitutively expressed in cells and commonly used as internal controls in Western blotting (e.g. GAPDH, β-actin, tubulin), it is not feasible to include a universal loading control that is applicable to both sample types. For instance, as shown in our results, β-actin, detected in whole-cell-lysate, was absent in the exosome lysate, confirming the absence of “contaminating” cellular material and the purity of the exosome samples. This challenge is well recognized in exosome research, as no well-established internal control proteins are available for exosome samples. To address this, we ensured equal loading by accurately quantifying protein concentrations (according to BCA assay) prior to electrophoresis. Additionally, for exosome samples, we followed standard practices by verifying exosome-specific protein markers as part of their qualitative characterization.]
Comments 5: [Western blot image does not have a loading marker.] Response 5: [Thank you for your comment regarding the inclusion of a loading marker in the Western blot image. We would like to clarify that a loading marker (Color Prestained Protein Standard, Broad Range (New England Biolabs) was included in the experiment, as also correctly mentioned in the results section. However, we inadvertently omitted the information about the protein marker in the Methods section.
Line 222 -page 6- We have revised the “CD9 and CD63 exosomal markers identification with Western Blot and Flow Cytometry analysis” section of the Methods, to include the details of the protein marker (mentioned as comment in the manuscript).
The experiments were conducted using PVDF membranes and HRP-conjugated antibodies, with detection performed via X-ray films to identify the protein bands. Due to this detection method, the loading marker is not visible in the final film-based image, as it is only visible on the PVDF membrane. Since no antibodies are bound to the protein bands of the marker, no fluorescence signals are detected on the film. To ensure equal loading, we carefully quantified protein concentrations for all samples prior to electrophoresis.]
Comments 6: [The figure 5 a miRNA fold change compared to control should be mentioned for representative miRNAs.] Response 6: [Thank you for your comment regarding the inclusion of miRNA fold change in Figure 5. The image presents the most significantly enriched miRNAs in A549 cell-derived exosomes, as determined using the calculation method described in Section 2.7 of the Methods (page 8). This analysis focuses on the enrichment of miRNAs within A549 cell-derived exosomes and is not a comparative or differential expression analysis. Therefore, fold change data are not applicable.]
Comments 7: [Few pictures specially enrichment analysis and network analysis are not high quality. Its hard to read them.] Response 7: [Thank you for your careful evaluation and feedback regarding the quality of the enrichment and network analysis images. In response to your comment, we have replaced some images with higher-quality versions to ensure they are clearer and easier to read. These images have been re-loaded. Replaced images: Figure 3, page 13 Figure 5, page 17 Figure 8, page 22 All other images were exported at the highest quality available. However, we understand that the complexity of some networks may require zooming in and careful observation to view all details clearly. We appreciate your suggestion, which has helped us improve the overall presentation of the manuscript.]
|
||
|
4. Response to Comments on the Quality of English Language |
||
|
Point 1: The quality of English does not limit my understanding of the research. |
||
|
Response 1: Thank you for your careful evaluation. |
||

Round 2
Reviewer 2 Report
Comments and Suggestions for Authors
Thank you for revising the manuscript. However the figure 2 formatting still needs attention. There is an overlap of figures.
Comments on the Quality of English LanguageEnglish language is fine.